

# A comparison of the mineral element content of 70 different varieties of pear fruit (*Pyrus ussuriensis*) in China

Chang Liu[1,2,*], Honglian Li[3,*], Aihua Ren[4], Guoyou Chen[5],
Wanjun Ye[6], Yuxia Wu[1], Ping Ma[1,7], Wenquan Yu[2] and Tianming He[1]

[1] College of Horticulture, Xinjiang Agricultural University, Urumqi, China
[2] Mudanjiang Branch, Heilongjiang Academy of Agricultural Sciences/Key Laboratory of Fruit Breeding and Cultivation in Cold Areas, Mudanjiang, Heilongjiang, China
[3] Institute of Pomology, Jilin Academy of Agricultural Sciences, Gongzhuling, Jilin, China
[4] Horticulture Branch, Heilongjiang Academy of Agricultural Sciences, Harbin, China
[5] Quality and Safety Institute of Agricultural Products, Heilongjiang Academy of Agricultural Sciences/Inspection and Testing Center for Quality of Cereals and Their Products (Harbin), Ministry of Agriculture and Rural Affairs, Heilongjiang, China
[6] Heilongjiang Academy of Agricultural Sciences, Harbin, China
[7] Bayin Guoleng Vocational and Technical College, Korla, China
* These authors contributed equally to this work.

Corresponding author
Tianming He, 1554272245@qq.com

## ABSTRACT

**Background:** *Pyrus ussuriensis* (Maxim.) is a unique pear tree that grows in northern China. The tree has strong cold resistance and can withstand low temperatures from $-30\,°C$ to $-35\,°C$. Due to its unique growth environment, its fruit is rich in minerals and has much higher levels of minerals such as K, Ca and Mg than the fruit of *Pyrus pyrifolia* (Nakai.) and *Pyrus bretschneideri* (Rehd.) on the market, and many say the ripe fruit tastes better than other varieties. A comprehensive analysis of the characteristics of mineral elements in the fruits of different varieties of *P. ussuriensis* will provide a valuable scientific basis for the selection, breeding and production of consumer varieties of *P. ussuriensis*, and provide a more complete understanding of nutritional differences between fruit varieties.
**Methods:** In this study, 70 varieties of wild, domesticated and cultivated species of *P. ussuriensis* from different geographical locations were compared. Targeting four main mineral elements and eight trace mineral elements contained in the fruit, the differences in mineral content in the peel and pulp of different varieties of *P. ussuriensis* were analyzed, compared and classified using modern microwave digestion ICP-MS.
**Results:** The mineral elements in the fruit of *P. ussuriensis* generally followed the following content pattern: K > P > Ca > Mg > Na > Al > Fe > Zn > Cu > Cr > Pb > Cd. The mineral element compositions in the peel and pulp of different fruits were also significantly different. The four main mineral elements in the peel were K > Ca > P > Mg, and K > P > Mg > Ca in the pulp. The mineral element content of wild fruit varieties was higher than that of cultivated and domesticated varieties. Correlation analysis results showed that there was a significant positive correlation between K, P and Cu in both the peel and pulp of *P. ussuriensis* fruit ($P < 0.01$). Cluster analysis results showed that the 70 varieties of *P. ussuriensis* could be divided into three slightly different categories according to the content of the peel or pulp. According to the contents of the fruit peel, these varieties were divided into: (1) varieties with high

Na, Mg, P, K, Fe and Zn content, (2) varieties with high Ca content and (3) varieties with medium levels of mineral elements. According to the fruit pulp content, these varieties were divided into: (1) varieties with high Mg, P and K content, (2) varieties with low mineral element content, and (3) varieties with high Na and Ca content. The comprehensive analysis of relevant mineral element content factors showed that 'SSHMSL,' 'QYL,' 'SWSL' and 'ZLTSL-3' were the best varieties, and could be used as the focus varieties of future breeding programs for large-scale pear production.

# INTRODUCTION

The pear belongs to the Rosaceae family and is one of the most important fruit trees in the world. Based on origin, pear varieties are generally divided into two categories: Asian pears and western pears (*Chen et al., 2018*). In China, the Asian pear is the main category of pear, with a cultivation history of more than 3,000 years, and is the third largest fruit grown, by volume, after citrus and apple. Pears are widely planted in China, mainly distributed in North China, Northeast China, Northwest China and the Yangtze River basin, with rich germplasm resources (*Dong et al., 2018*; *Li et al., 2022*). Previous research has been conducted on pear germplasm resources using agronomic and quality traits (*Zheng et al., 2022*; *Wu et al., 2018*; *Niu et al., 2019*; *Gong et al., 2020*), Relevant fruit research has found that there are many ascorbic acids, flavonoids, sugars, organic acids and minerals (K, P, Mg, Ca) and other bioactive substances that are beneficial to human health (*Hou et al., 2022*; *Li et al., 2018a*; *Adhikary et al., 2020*; *Sun et al., 2021*; *Che, Yamaji & Ma, 2018*). Both Chinese traditional medicine and international modern medicine believe that pears have an antioxidant effect and are capable of lowering blood pressure, moistening the lungs, resolving phlegm clearing heat and detoxification the body, Eating pears regular can help supplement the nutrition needed by the human body, and may even have medicinal value (*Wang et al., 2015*; *Li et al., 2014*; *Peng et al., 2023*; *Ogita et al., 2020*; *Bajwa et al., 2015*). With the increased enthusiasm for healthy, sustainable food and the acceleration of farming technology the importance of pear cultivation as an industry has increased in many places (*Zhang, Zhang & Gao, 2012*; *Li et al., 2018b*).

This increase in pear cultivation has led to higher fruit quality standards in the global market, especially the high-end market and recent research has focused on the development of more nutritional and better tasing fruit varieties. Wild apples in Xinjiang have been used to cultivate new varieties of (red pulp) apples with high-flavonoid content, and a new variety of pear 'Shannongsu' with good antioxidant quality, has been developed by crossing the unique 'Korla fragrant Pear' with the 'Dangshansu Pear'. These examples show the potential for innovative alongside the optimization of fruit resources (*Chen et al., 2022*).

Mineral element content is one way to measure fruit quality traits, as mineral elements are closely related to fruit size, pulp hardness, and soluble solids (*Mészáros et al., 2021*; *Maity et al., 2022*; *Sete et al., 2019*), and play important roles in fruit disease resistance,
storage resistance, and maintaining good quality and flavor (*Cui et al., 2020*; *Wei et al., 2017*). Mineral elements are also very important nutrients that are essential for healthy human growth and development (*Gorący et al., 2021*). If the human body lacks mineral Elements, it suffers from 'hidden hunger' (*Gödecke, Stein & Qaim, 2018*). Previous studies have shown that mineral elements play a crucial role in cell metabolism, biosynthesis and immune function of the human body by combining with proteins and other organic molecular functional groups (*Aranaz et al., 2020*; *Njoku et al., 2018*). For example, iron (Fe) is an essential component of hemoglobin, myoglobin and some functional enzymes (*McDevitt et al., 2020*); zinc (Zn) is a component of many important enzymes in the human body, and a lack of Zn lead to declines in immune function and can cause a variety of diseases (*Nadeem et al., 2020*). Many of the mineral elements needed by the human body cannot be synthesize on their own and need to be supplemented by the intake of fruits such as apples, pears, or grapes. Therefore, it is important to study the content of mineral elements in pears and other fruits for the selection of better functional varieties for fruit tree cultivation. *Pyrus ussuriensis* (Maxim.) is one of the main pear varieties cultivated in China (*Qiu et al., 2018*), and includes popular varieties such as: 'Nanguo Pear', 'Jinxiangshui pear', and 'Huagai pear'. It is mainly distributed in Heilongjiang, Jilin, Liaoning and other eastern regions of China. These areas are hot and rainy in the summer, which is conducive to vegetation growth. Winter in these areas is long, cold and dry, with large temperature difference between day and night causing the surface vegetation to form humus after long-term corrosion, which can evolve into rich soil with high organic content. Pears planted under these conditions are favored because of their good flavor and high nutritional value, and play a very important role in the cultivation of related fruit trees and fruit production.

There are many previous studies on the content of mineral elements in pear fruits, but these studies have mainly been carried out on *P. pyrifolia*, *P. bretschneideri* and *P. communis* L. (*Shen et al., 2019*; *Su et al., 2017*; *Kong et al., 2018*; *Saquet, Streif & Almeida, 2019*). The research reports that have been done on *P. ussuriensis* have only analyzed the mineral content of the 'Pingguo Pear' and 'Nanguo Pear.' These studies have found that K, N and Ca content in the fruits of different orchards can vary greatly (*Yan et al., 2022*; *Piao et al., 2018*), and that Mg and Ca content are much higher in *P. ussuriensis* than in *P. pyrifolia*. and *P. bretschneideri* (*Liu et al., 2022*).

Although previous work has provided valuable contributions to the understanding of the nutritional content of pears, because the sample sizes of varieties were too small in these previous studies, the mineral element content findings are not universal.This study selected 70 mature *P.ussuriensis* varieties from wild, domesticated and cultivated species from different geographical locations to better understand the distribution characteristics of material element content and the differences between the different varieties. Targeting four main mineral elements and eight trace mineral elements contained in *P. ussuriensis*, the differences in the mineral contents of the peel and pulp of different varieties of *P. ussuriensis* were analyzed, compared, classified using modern microwave digestion ICP-MS. The purpose of this study was to determine the content characteristics and regional distribution of mineral elements in different varieties of *P. ussuriensis* and screen

**Table 1  Basic information of *Pyrus ussuriensis* (Maxim.) varieties.**

| Code | Name | Type | Code | Name | Type | Code | Name | Type |
|---|---|---|---|---|---|---|---|---|
| P1 | SPL | Domesticated varieties | P25 | FXL | Domesticated varieties | P49 | CXL | Cultivated varieties |
| P2 | DXL | Domesticated varieties | P26 | QXSL | Domesticated varieties | P50 | WX | Cultivated varieties |
| P3 | XPXL | Domesticated varieties | P27 | QPHL | Domesticated varieties | P51 | NGL | Cultivated varieties |
| P4 | WXL | Domesticated varieties | P28 | QHL | Domesticated varieties | P52 | XXSL | Cultivated varieties |
| P5 | PDXL | Domesticated varieties | P29 | YBDJBL | Domesticated varieties | P53 | DN5L | Cultivated varieties |
| P6 | PDX-1 | Domesticated varieties | P30 | HLJHGL | Domesticated varieties | P54 | ZXSL | Cultivated varieties |
| P7 | PDX-2 | Domesticated varieties | P31 | TL | Domesticated varieties | P55 | JXSL | Cultivated varieties |
| P8 | HGL | Domesticated varieties | P32 | HPXL | Domesticated varieties | P56 | DML | Cultivated varieties |
| P9 | HLL | Domesticated varieties | P33 | BLXL | Domesticated varieties | P57 | TXL | Cultivated varieties |
| P10 | ML | Domesticated varieties | P34 | MTHL | Domesticated varieties | P58 | JTL | Cultivated varieties |
| P11 | QPCL | Domesticated varieties | P35 | AL | Domesticated varieties | P59 | SSDL | Wild resources |
| P12 | HXL | Domesticated varieties | P36 | HTL | Domesticated varieties | P60 | SSHMSL | Wild resources |
| P13 | LHHXSL | Domesticated varieties | P37 | TXDYXL | Domesticated varieties | P61 | SSTESL | Wild resources |
| P14 | HHGL | Domesticated varieties | P38 | SL | Domesticated varieties | P62 | SSKWSL | Wild resources |
| P15 | XCZL | Domesticated varieties | P39 | DWBKSL | Domesticated varieties | P63 | DNSL | Wild resources |
| P16 | CBZL | Domesticated varieties | P40 | DNGL | Cultivated varieties | P64 | HLSL | Wild resources |
| P17 | BLL | Domesticated varieties | P41 | FX | Cultivated varieties | P65 | SWSL | Wild resources |
| P18 | LDL | Domesticated varieties | P42 | YY351L | Cultivated varieties | P66 | ZLTSL-1 | Wild resources |
| P19 | HMXSX | Domesticated varieties | P43 | QYL | Cultivated varieties | P67 | ZLTSL-2 | Wild resources |
| P20 | SLGZL | Domesticated varieties | P44 | JXL | Cultivated varieties | P68 | ZLTSL-3 | Wild resources |
| P21 | JBL | Domesticated varieties | P45 | HJQL | Cultivated varieties | P69 | HEBSL-1 | Wild resources |
| P22 | XHTL | Domesticated varieties | P46 | LXL | Cultivated varieties | P70 | HEBSL-2 | Wild resources |
| P23 | DXSL | Domesticated varieties | P47 | DXS | Cultivated varieties | | | |
| P24 | YHL | Domesticated varieties | P48 | YY1L | Cultivated varieties | | | |

**Note:**
The 70 samples we studied were selected according to the main production areas of *Pyrus ussuriensis* (Maxim.) in northeast China. The geographical locations of the selected samples include five from Inner Mongolia, 28 from Heilongjiang, 34 from Jilin and three from Liaoning.

out the resources with high K, Ca and P content. These research results provide insights into maximizing the nutritional value of *P. ussuriensis* and contribute to the further development and utilization of *P. ussuriensis* resources in different regions. The results also provide valuable reference materials for the in-depth study of the molecular construction mechanism of mineral elements in plant cell structure, and the interactions and functions of various mineral elements.

# MATERIALS AND METHODS

## Sample collection and preparation

This study was carried out in the Fruit Tree Institute of Jilin Academy of Agricultural Sciences in 2021. There were 70 pear varieties tested, including 39 domesticated varieties, 19 cultivated varieties and 12 wild varieties (see Table 1). The sample fruit was collected from the National Field Genebank for Hardy Fruits (Gongzhuling City, Jilin Province) at the mature stage (based on the blackening of the seed color). The trees were grown in a flat

**Table 2 Reference parameters for microwave-assisted digestion.**

| Step | Power/w | Percentage/% | Heating-up time/min | Controlled temperature/°C | Duration/min |
|---|---|---|---|---|---|
| 1 | 600 | 100 | 8 | 100 | 5 |
| 2 | 600 | 100 | 5 | 150 | 5 |
| 3 | 600 | 100 | 8 | 190 | 15 |

**Note:**
Microwave digestion is divided into 3 steps, indicating the heating time, control temperature and duration of each step are detailed.

plot under standardized management, and the fruit was not bagged. A total of 20 pears were randomly picked from the same height around the crown of five 10-year-old trees of each variety, and 10 pears of similar same shape size and color were selected for analysis from the 20 sample fruits from each variety. These fruits were then immediately sent to the laboratory under fresh-keeping conditions where they were cleaned and dried. The peel and pulp of the pears were made into uniform paddles by the uniform paddling machine. The plastic sealed bags were separately packed and stored in cold storage for testing. Three replicates were made from each sample.

Inductively coupled plasma mass spectrometry was performed using the method outlined by *Yang et al. (2022)*. This method included: weighing 1.0–1.5g of the samples into the microwave digestion inner tank, adding 5 mL of nitric acid and 2 mL of hydrogen peroxide, covering and tightening the nut, and placing for 30 min. After a protective sleeve was installed, the samples were then pre-digested and placed symmetrically on the turntable, which was loaded with the classic method. The star/pause key was then pushed to start the digestion procedure (the digestion reference conditions are shown in Table 2). After the digestion procedure was completed, the instrument automatically entered the cooling process. When the indicated temperature in the instrument chamber was less than 50 °C, the digestive tube ware removed, and the nut in the fume hood was slowly unscrewed to relieve pressure. The digestive solution was then transferred with deionized water to a constant volume of 50 ml, mixed well for testing, and then the sample blank test was conducted.

## Instruments and reagents

The instruments used for samples preparation and analysis include: microwave digestion system (MARS 6, USA), ICP-MS inductively coupled plasma mass spectrometer (ICAP Q, USA), ultrapure water manufacturing system (Milli-Q, USA), and 6875D fully-automatic frozen grinder (SPEX Sample Prep, USA). The glassware and polytetrafluoroethylene digestion tank used in the experiment were soaked in 20% nitric acid for more than 24 h, and washed with ultra-pure water repeatedly, and then dried before use. The mixed standard of Na, Mg, Al, P, K, Ca, Cr, Fe, Cu, Zn, Cd and Pb (100 mg·L-1) was provided by the National Institute of Nonferrous Metals and Electronic Materials, and was prepared to the required concentration when used. The lithium (Li), cobalt (Co), indium (In), uranium (U) mixed mass spectrometry tuning solution (10 mg·L-1), rhodium (Rh) element internal standard solution (1,000 mg·L-1), nitric acid and hydrogen peroxide were MOS level.

**Table 3 Linear regression equations and correlation coefficients of the 12 mineral elements.**

| Element | Linear range | Standard curve equation | Correlation coefficient | Method detection limits |
|---|---|---|---|---|
| Na | 0–2,000 | f(x) = 7,738.4555 * x | R = 0.9998 | 2,500 |
| Mg | 0–2,000 | f(x) = 3,922.2351 * x | R = 0.9998 | 2,500 |
| Al | 0–2,000 | f(x) = 1,817.0022 * x | R = 0.9999 | 1,000 |
| P | 0–2,000 | f(x) = 185.2859 * x | R = 0.9981 | 1,000 |
| K | 0–2,000 | f(x) = 8,389.8404 * x | R = 0.9900 | 2,500 |
| Ca | 0–2,000 | f(x) = 427.4738 * x | R = 0.9879 | 2,500 |
| Cr | 0–500 | f(x) = 2,7376.5899 * x | R = 0.9998 | 25 |
| Fe | 0–2,000 | f(x) = 354.2781 * x | R = 0.9981 | 500 |
| Cu | 0–1,000 | f(x) = 4,068.2915 * x | R = 0.9999 | 100 |
| Zn | 0–2,000 | f(x) = 959.5661 * x | R = 0.9999 | 500 |
| Cd | 0–200 | f(x) = 1,592.8765 * x | R = 0.9999 | 2.5 |
| Pb | 0–500 | f(x) = 13,304.5313 * x | R = 0.9999 | 10 |

Note:
The appropriate concentration range of the standard working curve was selected based on the characteristic response value of 12 elements and the content conditions of the sample. Then, the standard curve equation and linear correlation coefficient were obtained.

## Preparation of the standard curve

ICP-MS has a wide dynamic linear range, high sensitivity and fast analysis speed. First, the appropriate concentration range of the standard working curve was selected based on the characteristic response value of 12 elements and the content conditions of the sample. Then, the standard curve equation and linear correlation coefficient were obtained, ICP-MS was then used to determine the content of the 12 elements in the samples (no less than 11 sample blank parallel) processed simultaneously with an unknown sample. The method detection limit test was then carried out, and the standard deviation of the response value of each element was calculated, The instrument detection limit was calculated by dividing three times the standard deviation by the slope of the standard curve. The detection limit of the method was calculated using the mass of the sample and constant volume (results shown in Table 3).

## Statistical analysis

The mean value (MEAN), standard deviation (SD) and coefficient of variation (CV) of the traits were calculated using Microsoft Excel 2017 and were statistically analyzed using IBM SPSS Statistics 22.0 software. The average amount of each element in different varieties of *P. ussuriensis* was analyzed using single factor analysis of variance and a Duncan multiple comparison at 5% and 1% significant levels. The correlation analysis used an equidistant Pearson similarity analysis, and Origin 2021b was used to draw a heat map. A principal component analysis was used for the factor analysis, defining the factors with a cumulative contribution rate greater than 80% as principal components, and then using these principal components to evaluate different *P. ussuriensis* resources. The clustering analysis was performed using the "analysis-classification-system clustering" process in the SPSS

**Table 4 Difference in mineral element content between the peel and the pulp of 70 *Pyrus ussuriensis* (Maxim.) varieties (*n* = 3, mg/kg).**

| Element | Peel | | | | Pulp | | | |
|---|---|---|---|---|---|---|---|---|
| | Average | Range | SD | CV(%) | Average | Range | SD | CV(%) |
| $^{39}$K | 1,838.06Aa | 885.26–3,209.00 | 547.9 | 29.81 | 1615.81Aa | 626.00–3,079.00 | 543.39 | 33.63 |
| $^{44}$Ca | 226.42Bb | 43.01–833.40 | 132.26 | 58.41 | 63.51Cd | 23.50–363.96 | 42.97 | 67.66 |
| $^{31}$P | 188.75Cc | 73.05–388.40 | 65.61 | 34.76 | 141.59Bb | 53.64–316.90 | 57.53 | 40.63 |
| $^{24}$Mg | 166.90Cc | 60.78–285.00 | 54.14 | 32.44 | 78.23Cc | 41.40–126.95 | 21.93 | 28.03 |
| $^{23}$Na | 9.40Dd | 4.67–51.46 | 7.25 | 77.13 | 5.67De | 2.45–14.21 | 2.28 | 40.17 |
| $^{27}$Al | 10.18Dd | 3.11–47.35 | 6.71 | 65.9 | 4.22De | 0.62–15.13 | 2.43 | 57.5 |
| $^{57}$Fe | 9.23Dd | 3.24–27.83 | 5.06 | 54.83 | 4.27De | 1.83–8.59 | 1.55 | 36.3 |
| $^{66}$Zn | 1.44Dd | 0.67–4.39 | 0.64 | 44.71 | 0.69De | 0.27–1.46 | 0.26 | 38.13 |
| $^{63}$Cu | 1.27Dd | 0.38–2.78 | 0.55 | 42.96 | 0.57De | 0.12–1.23 | 0.23 | 40.91 |
| $^{52}$Cr | 0.22Dd | 0.08–0.47 | 0.14 | 65.73 | 0.11De | 0.04–0.20 | 0.03 | 28.23 |
| $^{208}$Pb | 0.0435Dd | 0.0140–0.0841 | 0.0137 | 31.52 | 0.0226De | 0.0110–0.0512 | 0.0091 | 40.1 |
| $^{111}$Cd | 0.0054Dd | 0.0020–0.0111 | 0.0022 | 41.09 | 0.0038De | 0.0010–0.0161 | 0.0023 | 59.36 |

Notes:
Different lowercase letters in the same column indicate significant differences at the 0.05 level, and different capital letters indicate significant differences at the 0.01 level. The mean value (MEAN), standard deviation (SD), range of variation (Range) and coefficient of variation (CV) of the 12 elements were calculated. The average content of each element in different varieties of *P. ussuriensis* was analyzed by single factor analysis of variance and Duncan multiple comparison at 5% and 1% significant levels.

software, and the hierarchical clustering diagram was drawn using the square Euclidean distance in the Origin 2021b software.

# RESULTS

## Difference analysis of mineral element content

The content range, average values, standard deviations and coefficients of variation of the 12 mineral elements in the 70 *P. ussuriensis* varieties are listed in Table 4. The results showed that the average content of different mineral elements in fruit followed the following order from highest to lowest: K >P > Ca > Mg > Na > Al > Fe > Zn > Cu > Cr > Pb > Cd. The K content accounted for 79.10% of the total mineral element content in the fruit, which is about 10, 14 and 12 times of the content of three other major elements P, Mg and Ca, respectively. K, P, Mg and Ca content in the fruit peel and pulp were much larger than the content of the eight trace elements, which had a relative range between 0.47 and 4,132.51. The correlation analysis of variance showed that there were significant differences in the content order of K, P, Mg and Ca in the peel and pulp, with the mineral content in the peel following a K > Ca > P > Mg order, and the mineral content in the pulp following a K > P > Mg > Ca order. The content of trace elements were Na, Al, Fe, Zn, Cu, Cr, Pb, Cd (in this order) and was not significantly different between the fruit peel and the fruit pulp. The largest coefficient of variation in the peel was Na, followed by Al and Cr, and the largest coefficient of variation in the pulp was Ca, followed by Cd and Al. These results showed that the Na content in peel and the Ca content in the pulp of different varieties of *P. ussuriensis* varied greatly. The mineral element content in the peel of different varieties of *P. ussuriensis* varied greatly, as shown in Fig. 1, with K content being the highest, and Cd content being the lowest of the measured elements. Ca content in the

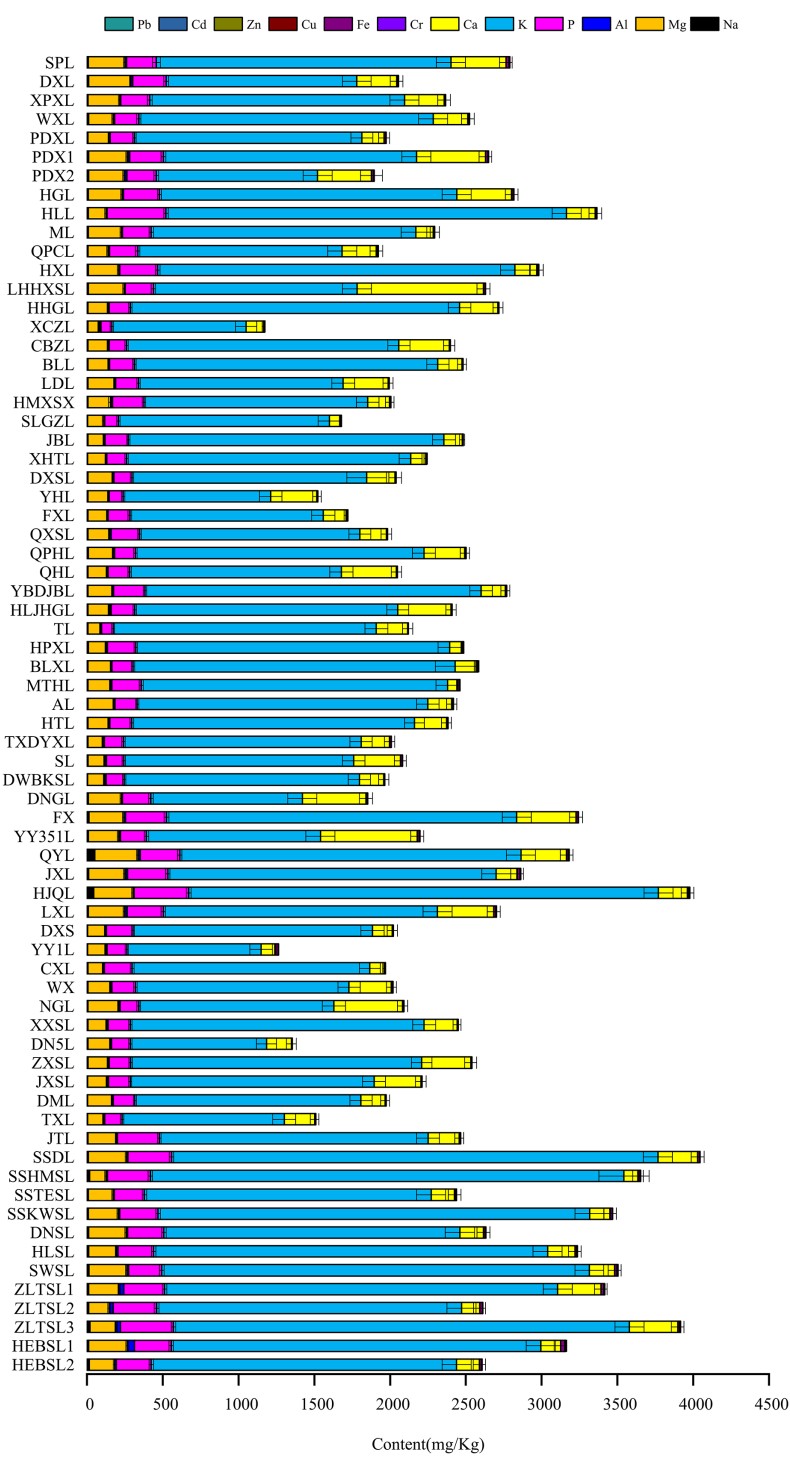

**Figure 1 Content of mineral elements in the peel of *Pyrus ussuriensis* (Maxim.).** The vertical axis is the 70 *P. ussuriensis* varieties, the horizontal axis is the total content of mineral elements in the peel of each variety, and the different colors of the columnar superposition graph are the total content of each mineral element.

different varieties of *P. ussuriensis* had the largest range, from 43.01 to 833.40 mg/kg, The K content range was the smallest, ranging from 885.26 to 3,209.00 mg/kg in different varieties. Of the 70 cultivated, domesticated and wild varieties included in the study, the *P. ussuriensis* varieties with the highest Ca content were 'YY351L' (638.50 mg/kg), 'LHHXSL' (833.40 mg/kg) and 'ZLTSL-3' (318.54 mg/kg). The *P. ussuriensis* varieties with the lowest Ca content were 'DXS' (43.01 mg/kg), 'MTHL' (64.05 mg/kg) and 'SSHMSL' (92.60 mg/kg). The varieties of *P. ussuriensis* with the highest K content were 'HJQL' (3,099.00 mg/kg), 'HLL' (2,643.00 mg/kg) and 'SSDL' (3,209.00 mg/kg), and the varieties with the lowest K content were 'YY1L' (886.23 mg/kg), 'XCZL' (885.26 mg/kg) and 'SSTESL' (1,892.00 mg/kg).

Levels of the different mineral elements in the pulp of *P. ussuriensis* varieties are shown in Fig. 2, with K having the highest content and Cd having the lowest content. Al had the largest content range between varieties, ranging from 0.62–15.13 mg/kg. The Mg content range was the smallest, from 41.40–126.95 mg/kg. Of the 70 tested *P. ussuriensis* varieties, the varieties with the highest Al content were 'QYL' (15.13 mg/kg), 'QHL' (7.60 mg/kg) and 'ZLTSL-2' (5.53 mg/kg), and the varieties with the lowest Al content were 'FX' (0.78 mg/kg),'XPXL' (0.62 mg/kg) and 'DNSL' (1.44 mg/kg). The *P. ussuriensis* varieties with the highest Mg content were 'NGL' (124.39 mg/kg), 'YBDJBL' (126.95 mg/kg) and 'SWSL' (125.70 mg/kg), and the varieties of *P. ussuriensis* with the lowest Mg content were 'CXL' (45.00 mg/kg). 'HXL' (41.40 mg/kg) and 'ZLTSL-2' (54.60 mg/kg).

The 70 varieties of *P. ussuriensis* tested in this study were then classified into wild, cultivated or domesticated varieties. The results showed that although the Na and Cu content in the wild varieties were not significantly different from the Na and Cu content in the cultivated varieties, Mg, P, K, Fe, Cr and Zn content in the wild varieties were significantly higher than in the cultivated and domesticated varieties. Ca content in the cultivated varieties was significantly higher than in the wild and domesticated varieties. These results show that the domesticated varieties have no clear advantages in mineral element content. The comprehensive analysis of the mineral element content in the peel and pulp of these *P. ussuriensis* varieties found that 'ZLTSL-3' had the highest mineral element content, followed by 'SSKWSL' and 'SSHMSL,' and that overall, the levels of mineral elements in fruits of wild varieties was higher than in the fruit of the cultivated and domesticated varieties.

## Correlation analysis of mineral element content

Heat maps have been widely used to analyze, the nutritional quality of fruit in recent years because this method intuitively shows experimental results with a gradual blue-red band. The analysis results of this study show that there were significant or extremely significant correlations between mineral elements in the tested fruits. This indicates that there is a complex interaction between the cations of these mineral elements and the amino acid residues of plant cell protease, the phosphate and carboxylate anions both inside and outside the cell membrane, and the electrolytes of these mineral elements. These interactions are likely also impacted by factors such as growth environment and growth stage.

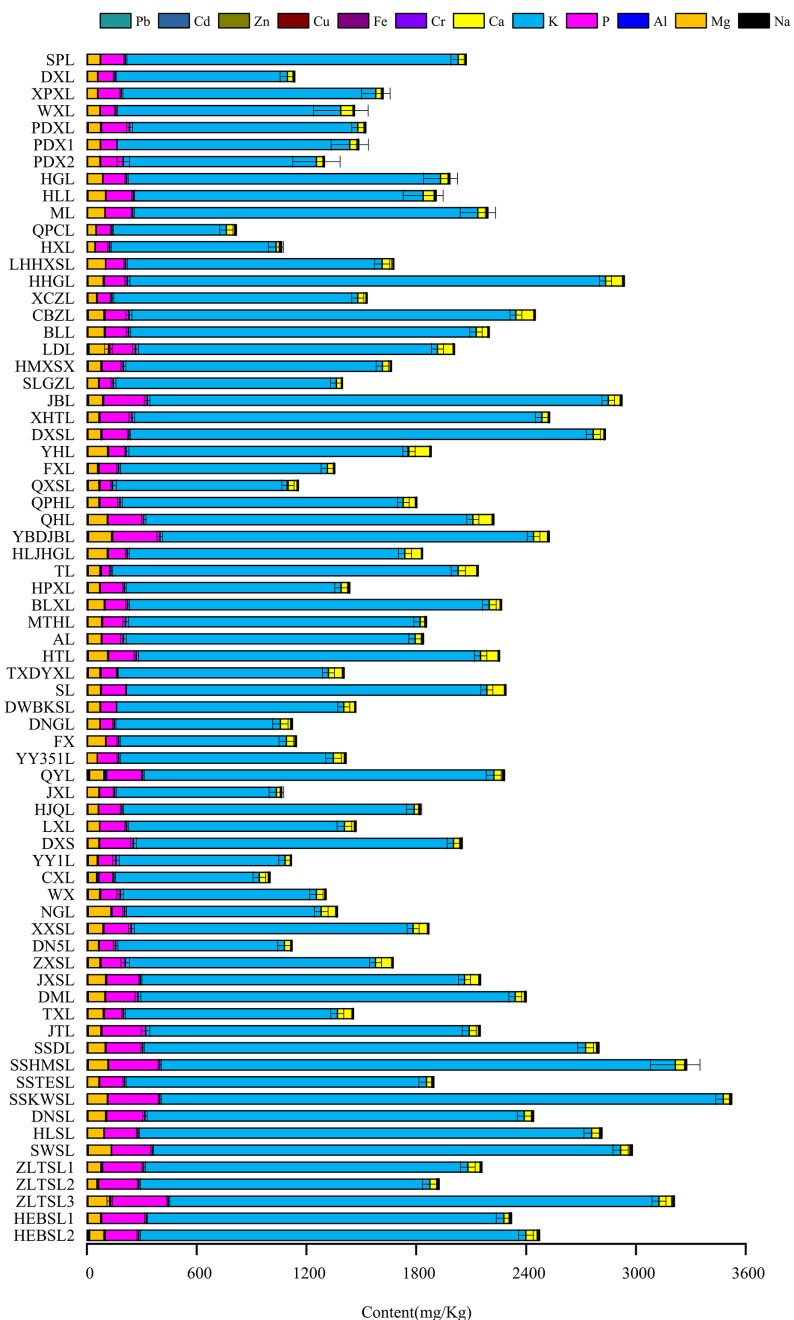

**Figure 2 Content of mineral elements in the pulp of *Pyrus ussuriensis* Maxim..** The vertical axis is 70 *P.ussuriensis* resources, the horizontal axis is the total content of mineral elements in the pulp of each resource, and the different colors of the columnar superposition graph are the content of mineral elements of each element.

Figure 3 shows that among the 12 mineral elements contained in the fruit peels of the 70 *P. ussuriensis* varieties, there were 33 pairs with an extremely significant positive correlation (*P* < 0.01). The Na contained in the peel had a very significant positive correlation with P and Cr, and their correlation coefficients were 0.541 and 0.514, respectively. Mg, P, Cr, Fe, Cu and other, elements contained in the fruit peel samples were

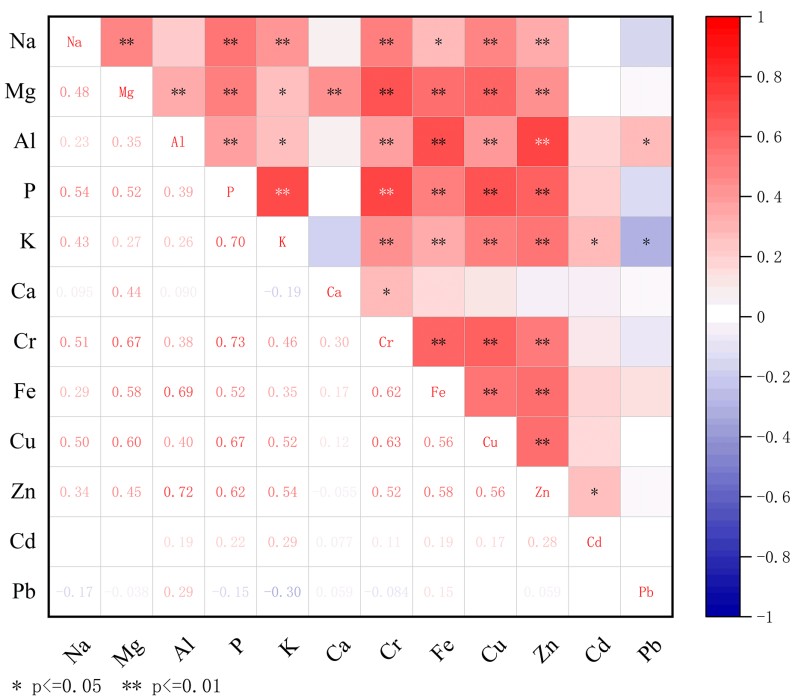

**Figure 3 Correlation of mineral elements in the peel of *Pyrus ussuriensis* (Maxim.).** Heat map of the content of 12 mineral elements in the peel, analyzed by correlation, and drawn using Origin 2021b software. The red and blue in the heat map indicate the degree of association between mineral elements, with red representing a positive correlation and blue representing a negative correlation. There are 33 pairs with extremely significant positive correlations ($P < 0.01$), and eight pairs with significant positive correlations ($P < 0.05$).

significantly positively correlated with each other, with correlation coefficients above 0.5. The correlation between P and Cr was relatively high, with a correlation coefficient of 0.729, indicating that the varieties with high P content in the peel were also likely to have high Cr content in the peel. Al was positively correlated with Fe and Zn, with correlation coefficients of 0.689 and 0.722, respectively. There was also a very significant positive correlation between P and K, P and Zn, K and Cu, K and Zn and Cu and Zn, with correlation coefficients of 0.702, 0.625, 0.517, 0.544, 0.563, respectively. Na and Fe, Na and K, Mg and Al, Mg and Cd, Al and Pb, Zn and Cd were all significantly positively correlated. Although there was a very significant positive correlation between Na and Mg, Na and K, Cu and Zn, Mg and Al, Ca and Zn, Al and P, Cr and Cu, K and Cr, and K and Fe, their correlation coefficients were relatively low. There was a significant negative correlation between K and Pb, with a correlation coefficient of −0.301, this abnormal result shows that in *P. ussuriensis* varieties with high levels of K, the Pb content was relatively low.

Figure 4 shows that among the 12 mineral elements studied in the fruit pulp of *P. ussuriensis* varieties, there were 30 pairs ($P < 0.01$) with extremely significant positive correlations, Na and Al, Na and Cr, Mg and K, Mg and Zn, Zn and Pb, had extremely significant positive correlations, with correlation coefficients of 0.662, 0.573, 0.517, 0.593 and 0.620, respectively. In addition, P had significant positive correlations with K, Fe, Cu and Zn. The correlation between P and K was high, with a correlation coefficient of 0.760,

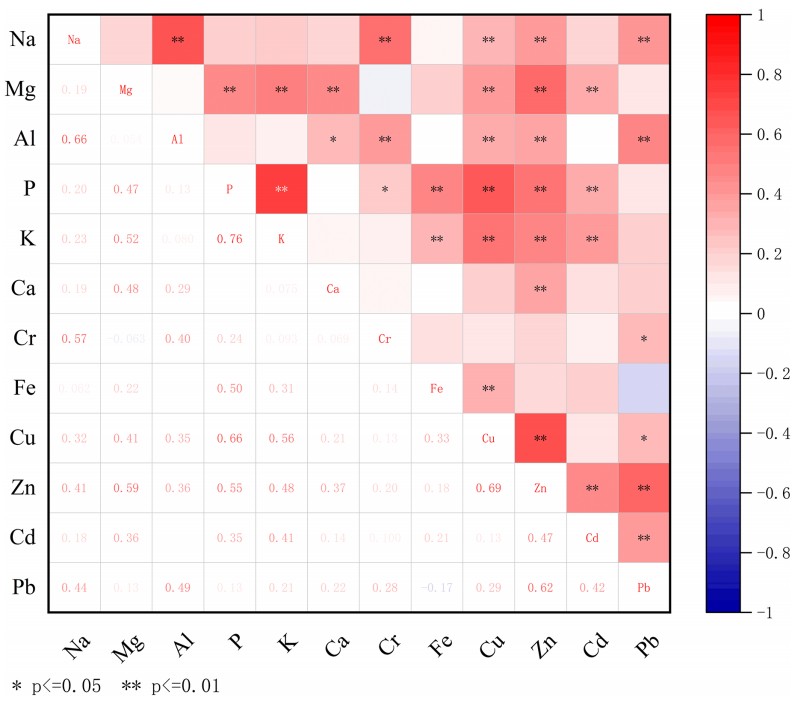

* p<=0.05  ** p<=0.01

**Figure 4 Correlation of mineral elements in the pulp of *Pyrus ussuriensis* (Maxim.).** Heat map of the content of 12 mineral elements in the pulp, analyzed by correlation, and drawn using Origin 2021b software. The red and blue in the heat map indicate the degree of association between mineral elements, with red representing a positive correlation and blue representing a negative correlation. There are 30 pairs with extremely significant positive correlations ($P < 0.01$), and four pairs with significant positive correlations ($P < 0.05$).

indicating that varieties with a high P content in fruit pulp also had high K content. Although there was a significant positive correlation between Al and Ca, P and Cr, and Cu and Pb, their correlation coefficients were relatively low. Differing from the fruit peel results, there was no significant negative correlation between any mineral elements in the fruit pulp.

## Factor analysis of mineral element content in *P. ussuriensis*

After standardization, this study conducted a factor analysis of the 12 mineral elements in 70 varieties of *P. ussuriensis*, mainly extracting common factors through a principal component analysis. The results found in Table 5, show that the cumulative variance contribution rate of the first five common factors selected in this study was greater than 80% of the total detected elements, indicating that these 12 mineral Elements accounted for the majority of the variance between these varieties.

Among the relevant factors of the fruit peel shown in Table 5, F1 included six mineral elements: Na (factor load: 0.843), P (factor load: 0.768), Cu (factor load: 0.744), Cr (factor load: 0.671), K (factor load: 0.620), and Mg (factor load: 0.565). F2 mainly contained three mineral elements: Al (factor load: 0.889), Fe (factor load: 0.796) and Zn (factor load: 0.773). F3, F4 and F5 were respectively related to Ca (factor load: 0.934), Pb (factor load: 0.953) and Cd (factor load: 0.969), Because the factor load was positive and in positive

**Table 5 Factor analysis of mineral element content.**

| Element | Peel | | | | | Element | Pulp | | | | |
|---|---|---|---|---|---|---|---|---|---|---|---|
| | F1 | F2 | F3 | F4 | F5 | | F1 | F2 | F3 | F4 | F5 |
| Na | 0.843 | 0.006 | 0.058 | −0.016 | −0.081 | Cu | 0.872 | 0.227 | 0.17 | −0.148 | −0.029 |
| P | 0.768 | 0.39 | −0.032 | −0.185 | 0.176 | P | 0.836 | 0.113 | −0.026 | 0.194 | 0.34 |
| Cu | 0.744 | 0.371 | 0.11 | 0.056 | 0.097 | K | 0.787 | 0.007 | 0.031 | 0.308 | 0.152 |
| Cr | 0.671 | 0.42 | 0.35 | −0.135 | 0.007 | Zn | 0.669 | 0.291 | 0.374 | 0.336 | −0.224 |
| K | 0.62 | 0.293 | −0.314 | −0.375 | 0.313 | Na | 0.174 | 0.836 | 0.123 | 0.093 | −0.035 |
| Mg | 0.565 | 0.376 | 0.561 | −0.059 | −0.108 | Cr | −0.025 | 0.817 | −0.112 | 0.145 | 0.269 |
| Al | 0.114 | 0.889 | 0.025 | 0.197 | 0.068 | Al | 0.174 | 0.794 | 0.186 | −0.141 | −0.238 |
| Fe | 0.305 | 0.796 | 0.232 | 0.043 | 0.036 | Ca | −0.009 | 0.174 | 0.922 | 0.02 | −0.045 |
| Zn | 0.396 | 0.773 | −0.131 | −0.031 | 0.187 | Mg | 0.493 | −0.101 | 0.679 | 0.249 | 0.119 |
| Ca | 0.028 | 0 | 0.934 | 0.043 | 0.078 | Cd | 0.197 | 0.053 | 0.122 | 0.92 | 0.038 |
| Pb | −0.105 | 0.174 | 0.007 | 0.953 | 0.021 | Fe | 0.342 | 0.086 | 0.035 | 0.091 | 0.787 |
| Cd | 0.046 | 0.123 | 0.058 | 0.013 | 0.969 | Pb | 0.246 | 0.483 | 0.073 | 0.454 | −0.587 |
| Variance contribution (%) | 43.173 | 13.444 | 12.006 | 8.132 | 5.257 | Variance contribution (%) | 36.852 | 17.236 | 11.537 | 8.772 | 7.135 |
| Cumulative variance contribution (%) | 43.173 | 56.617 | 68.623 | 76.755 | 82.011 | Cumulative variance contribution (%) | 36.852 | 54.088 | 65.624 | 74.396 | 81.531 |

**Note:**
The factor analysis of 12 mineral elements in the peel and pulp of 70 pear varieties, mainly extracting common factors through principal component analysis. The results show that the cumulative variance contribution rate of the first five common factors selected in this study was greater than 80% of the total detected elements, indicating that these 12 mineral elements accounted for the majority of the variance between these varieties.

distribution, the higher the factor score, the higher the corresponding mineral element content. Since the contribution of more than 50% of the total variance came from the sum of the two main factors, F1 and F2, the nine elements, Na, P, Cu, Cr, K, Mg, Al, Fe and Zn, can be defined as the characteristic mineral elements of the *P. ussuriensis* fruit peel.

Among the relevant factor of the fruit pulp shown in Table 5, F1 included four mineral elements: Cu (factor load: 0.872), P (factor load: 0.836), K (factor load: 0.787) and Zn (factor load: 0.669). F2 mainly contained three mineral elements: Na (factor load 0.836), Cr (factor load 0.817) and Al (factor load 0.794). F3, F4 and F5 are respectively related to Ca (factor load 0.922), Mg (factor load 0.679), Cd (factor load 0.920), Fe (factor load 0.787) and Pb (factor load −0.587), respectively. The factor load of Pb was negative, meaning the higher the Pb factor score, the lower the corresponding Pb content. Because more than 50% of the total variance come from the sum of the two main factors, F1 and F2, the seven elements, Cu, P, K, Zn, Na, Cr and Al can be defined as the characteristic mineral elements of *P. ussuriensis* fruit pulp.

Using the ratio of variance contribution rate of each principal component to cumulative variance contribution rate as the weight, this study established a comprehensive score modeld as follows:

*Peel: $Z_i = 0.4317Z(i, 1) + 0.1344Z(i, 2) + 0.1201Z(i, 3) + 0.0813Z(i, 4) + 0.0526Z(i, 5)$;*
*Pulp: $Z_i = 0.3685Z(i, 1) + 0.1724Z(i, 2) + 0.1154Z(i, 3) + 0.0877Z(i, 4) + 0.0714Z(i, 5)$*

Where, Zi is the comprehensive score; Z (i, 1), Z (i, 2), Z (i, 3), Z (i, 4), Z (i, 5) are the score of the five principal components; and i is the identifying number of the *P.ussuriensis* variety (1–70).

This formula can be used to calculate the sequence rule of the comprehensive component and content of mineral elements in *P. ussuriensis* fruit,which is helpful in identifying the most nutritious varieties. The higher the comprehensive component scores, the better the comprehensive mineral nutrition of the selected *P. ussuriensis* fruit variety (see Table 6 for more detailed results of different varieties). Among the 70 varieties of pears studied, the 15 varieties with the highest mineral element content in the peel were: 'SPL', 'FX', 'PDX-1', 'SSDL', 'HJQL', 'LXL', 'SSHMSL', 'PDX-2', 'QYL', 'SWSL', 'ZLTSL-2', 'JXL', 'ZLTSL-1', 'ZLTSL-3' and 'HEBSL-1'. The 15 fruit varieties with the highest mineral element content in the pulp were: 'HTL', 'SWSL', 'DXS', 'DML', 'JTL', 'ZLTSL-2', 'SSHMSL', 'CXL', 'LDL', 'YBDJBL', 'ZLTSL-3','HEBSL-2', 'QHL', 'JBL' and 'QYL'. According to the comprehensive data of the mineral element content in both the fruit peel and pulp, the best *P. ussuriensis* varieties were 'SSHMSL', 'QYL', 'SWSL' and 'ZLTSL-3'.

## Cluster analysis of mineral elements

In this experiment, the mineral element content of the fruit peel and pulp of 70 *P. ussuriensis* varieties were analyzed by cluster analysis (results are shown in Fig. 5). According to the content characteristics of the main mineral elements in the fruit peel, these 70 varieties of *P. ussuriensis* were divided into the following three categories:

Class I: Varieties with high Na, Mg, Al, P, K, Fe and Zn content in the peel. Nine varieties fell into this category, including 'HLL,' 'HJQL' and 'SSDL.' Na, Mg, Al, P, K, Fe and Zn content in these varieties were very high, with average values of 15.99, 193.58, 13.44, 293.96, 2882.81, 3.00 and 2.16 mg/kg, respectively.

Class II: Varieties with high Ca content in the peel. A total of 26 varieties fell into this category, including 'DXL,' 'PDXL' and 'QPCL.' The Ca content in these varieties was very high, with an average value of 257.35 mg/kg.

Class III: Varieties with medium element content in the peel. Half (35) of the varieties such fit into this category including 'SPL', 'XPXL' and 'WXL', The Na, Mg, Al, P, K, Fe and Zn content in these varieties were higher than in the varieties in Class II with average values of 9.59, 165.97, 10.43, 193.49, 1992.96, 8.95 and 1.50 mg/kg, respectively. The Ca content was also significantly lower, with an average of 196.97 mg/kg.

There were no significant differences in Cr, Cu, Cd and Pb content in the fruit peel between the three groups. Average Cr levels in the three groups were 0.40, 0.16 and 0.21 mg/kg, respectively, average Cu levels were 1.97, 1.06 and 1.28 mg/kg, respectively.

Average Cd levels were 0.0073, 0.0050 and 0.0052 mg/kg, respectively, and the average Pb levels were 0.0323, 0.0484 and 0.0435 mg/kg, respectively.

Similarly, according to the content characteristics of the main mineral elements of the fruit pulp these 70 varieties of *P. ussuriensis* were divided into three categories (results are shown in Fig. 6).

**Table 6 Principal component scores and ranking of 70 _Pyrus ussuriensis_ (Maxim.) varieties.**

| Code | Peel | | Pulp | |
|------|------|------|------|------|
| | Synthesis score | Ranking | Synthesis score | Ranking |
| P1 | 0.92 | 15 | −0.66 | 52 |
| P2 | 0.02 | 26 | −1.6 | 70 |
| P3 | −0.44 | 41 | −1.43 | 68 |
| P4 | −0.63 | 51 | −1.04 | 63 |
| P5 | −0.44 | 40 | −0.16 | 43 |
| P6 | 1.1 | 13 | −0.86 | 57 |
| P7 | 1.32 | 8 | −0.93 | 58 |
| P8 | 0.25 | 24 | −0.99 | 60 |
| P9 | 0.38 | 21 | −0.23 | 46 |
| P10 | −0.06 | 30 | −0.75 | 55 |
| P11 | −0.51 | 44 | −1.02 | 62 |
| P12 | 0.34 | 22 | −1.11 | 65 |
| P13 | 0.22 | 25 | −1.09 | 64 |
| P14 | −0.63 | 50 | 0.61 | 17 |
| P15 | −1.29 | 68 | −0.66 | 51 |
| P16 | −0.59 | 48 | 0.53 | 20 |
| P17 | −1.01 | 61 | 0.25 | 28 |
| P18 | −0.79 | 54 | 1.24 | 7 |
| P19 | −0.05 | 29 | 0.31 | 27 |
| P20 | −1.33 | 69 | −0.97 | 59 |
| P21 | −0.61 | 49 | 1.92 | 2 |
| P22 | −1.19 | 66 | −0.07 | 40 |
| P23 | −1.04 | 63 | −0.03 | 36 |
| P24 | −1.13 | 64 | 0.13 | 31 |
| P25 | −0.66 | 53 | −0.03 | 35 |
| P26 | −0.16 | 33 | −0.14 | 42 |
| P27 | −0.2 | 35 | −0.31 | 48 |
| P28 | −1.18 | 65 | 1.5 | 3 |
| P29 | −0.45 | 43 | 1.27 | 6 |
| P30 | −0.09 | 31 | −0.07 | 39 |
| P31 | −1.37 | 70 | 0.36 | 23 |
| P32 | −0.19 | 34 | 0.22 | 29 |
| P33 | −0.86 | 57 | 0.09 | 32 |
| P34 | −0.39 | 38 | −0.14 | 41 |
| P35 | −0.9 | 58 | −0.67 | 53 |
| P36 | −0.56 | 47 | 0.63 | 15 |
| P37 | −0.54 | 46 | 0.13 | 30 |
| P38 | −0.54 | 45 | −0.18 | 44 |
| P39 | −0.96 | 59 | −0.41 | 49 |
| P40 | −0.37 | 37 | −1.39 | 67 |

(Continued)

| Code | Peel | | Pulp | |
|------|------|------|------|------|
| | **Synthesis score** | **Ranking** | **Synthesis score** | **Ranking** |
| P41 | 0.92 | 14 | −1.44 | 69 |
| P42 | 0.81 | 16 | −1.01 | 61 |
| P43 | 1.41 | 7 | 1.93 | 1 |
| P44 | 1.74 | 4 | −1.3 | 66 |
| P45 | 1.16 | 11 | −0.24 | 47 |
| P46 | 1.19 | 10 | −0.69 | 54 |
| P47 | −1.23 | 67 | 0.75 | 13 |
| P48 | −0.64 | 52 | −0.07 | 38 |
| P49 | −0.45 | 42 | 1.03 | 8 |
| P50 | −0.22 | 36 | −0.2 | 45 |
| P51 | −0.03 | 28 | 0.38 | 22 |
| P52 | −0.44 | 39 | 0.33 | 25 |
| P53 | −1.03 | 62 | −0.05 | 37 |
| P54 | −0.8 | 55 | 0.04 | 34 |
| P55 | −0.99 | 60 | 0.49 | 21 |
| P56 | −0.15 | 32 | 0.76 | 12 |
| P57 | −0.81 | 56 | 0.31 | 26 |
| P58 | 0.29 | 23 | 0.78 | 11 |
| P59 | 1.15 | 12 | 0.61 | 16 |
| P60 | 1.28 | 9 | 0.84 | 9 |
| P61 | 0 | 27 | −0.86 | 56 |
| P62 | 0.72 | 20 | 0.34 | 24 |
| P63 | 0.8 | 17 | 0.07 | 33 |
| P64 | 0.73 | 19 | −0.48 | 50 |
| P65 | 1.47 | 6 | 0.66 | 14 |
| P66 | 2.06 | 3 | 0.56 | 18 |
| P67 | 1.64 | 5 | 0.82 | 10 |
| P68 | 2.23 | 2 | 1.39 | 5 |
| P69 | 3.04 | 1 | 0.53 | 19 |
| P70 | 0.76 | 18 | 1.47 | 4 |

**Note:**
Principal component score, calculated using the ratio of variance contribution rate of each principal component to cumulative variance contribution rate as the weight. The higher the comprehensive composition score, the better the comprehensive mineral nutrition of the selected *P. ussuriensis* fruit variety.

Class I: Varieties with high Mg, P and K content in the pulp. Nine varieties fell into this category, including 'HHGL,' 'JBL' and 'SWSL.' Mg, P and K content in these varieties were very high, with average values of 97.51, 221.84 and 2634.31 mg/kg respectively.

Class II: Varieties with low levels of all mineral content in the pulp. This category consisted of 24 varieties, including 'DXL', 'HXL' and 'FX'.
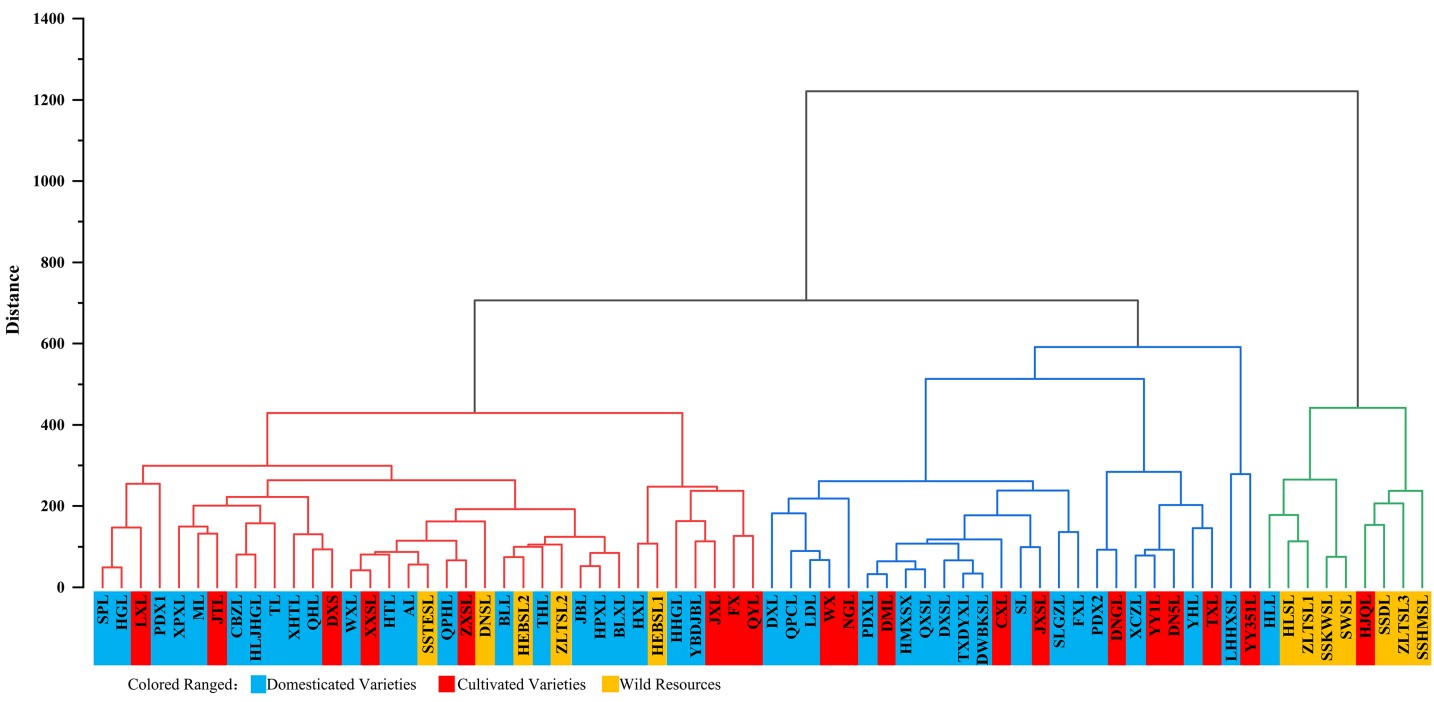

**Figure 5 Cluster analysis of the fruit peel of 70 *Pyrus ussuriensis* (Maxim.) varieties.** The hierarchical clustering diagram, drawn using the Origin 2021b software. According to the content of mineral elements in the fruit peel, the 70 *P. ussuriensis* varieties were divided into three groups. Different colors are used to represent different groups. The first group includes nine varieties and is marked in green, the second group includes 26 varieties and is marked in blue, and the third group includes 35 varieties and is marked in red.

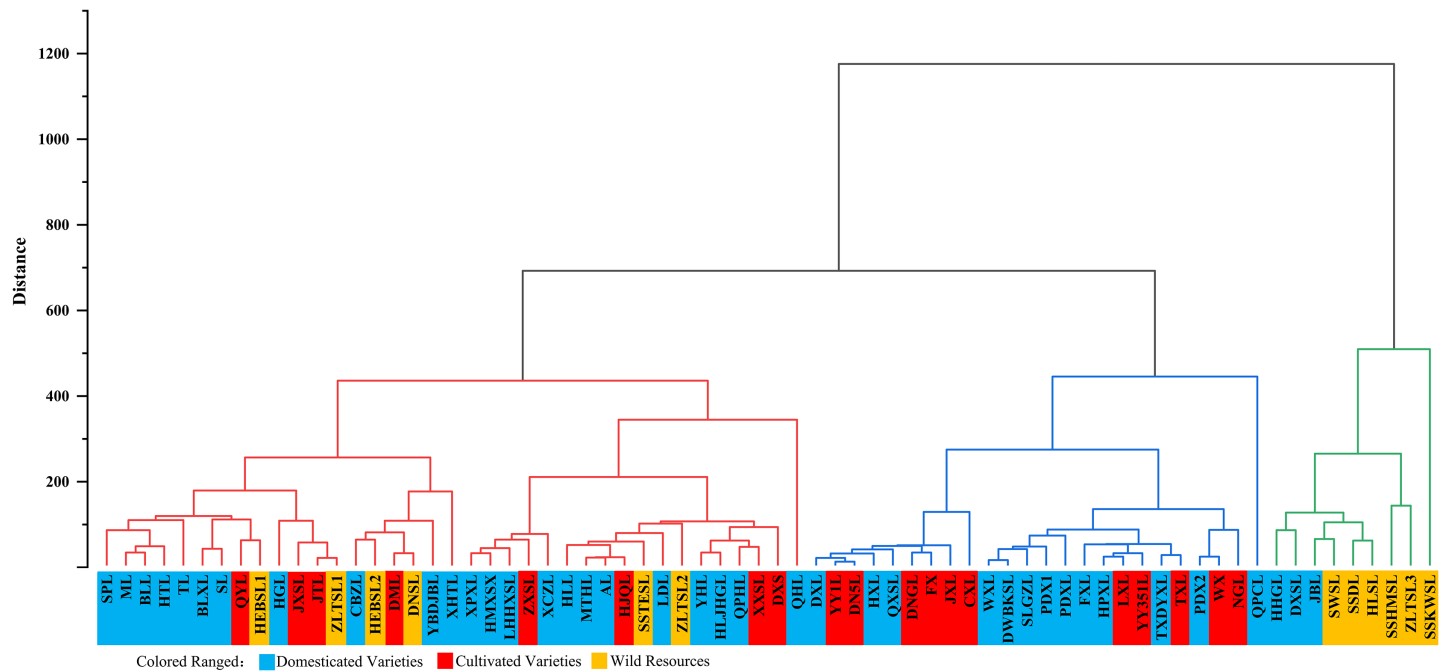

**Figure 6 Cluster analysis of the fruit pulp of 70 *Pyrus ussuriensis* (Maxim.) varieties.** The hierarchical clustering diagram, drawn using the Origin 2021b software. According to the content of mineral elements in the fruit pulp, the 70 *P. ussuriensis* varieties were divided into three groups. Different colors are used to represent different groups. The first group includes nine varieties and is marked in green, the second group includes 24 varieties and is marked in blue, and the third group includes 37 varieties and is marked in red.

Class III: Varieties with high Na, Al and Ca content in the pulp. A total of 37 varieties fell into this category, including 'SPL', 'ML' and 'BLL'. The average Na, Al and Ca content in these varieties were 6.12, 4.67 and 72.72 mg/kg, respectively.

There were no significant differences in Cr, Fe, Cu, Zn, Cd and Pb levels in the fruit pulp between the three groups. The average Cr content in the three groups was 0.11, 0.10 and 0.10 mg/kg, respectively; the average Fe content was 4.20, 3.91 and 5.51 mg/kg, respectively; the average Cu content was 0.60, 0.43 and 0.83 mg/kg, respectively; the average Zn content was 0.71, 0.56 and 0.96 mg/kg, respectively; the average Cd content was 0.0038, 0.0031 and 0.0058 mg/kg, respectively; and the average Pb content was 0.0223, 0.0210 and 0.0280 mg/kg, respectively.

## DISCUSSION

### Mineral element content of *P. ussuriensis* fruit

Mineral element content plays an important role in judging fruit quality and in maintaining the normal physiological activities of the human body (*Bai, Shen & Huang, 2021*; *Huang et al., 2022*; *Cao et al., 2018*; *Choi & Ha, 2013*; *Renna et al., 2018*). The coverage and sample size of previous *P. ussuriensis* research samples are relatively small (*Zhang et al., 2022a*).

The results of this study showed that the main element in all 70 *P. ussuriensis* fruits tested was K accounting for 79.10% of the total amount of 12 mineral elements studied. The average K content was about 10, 14 and 12 times that of the other three major elements P, Mg and Ca, respectively. Because its mobility is less bound by neighboring anions, the mobility of monovalent K cations in plants is significantly higher than that of divalent Ca and Mg cations. Moreover, in fruit cell tissue, it is easier to concentrate on the most active and necessary biochemical activities (*Dróżdż, Šéžienė & Pyrzynska, 2018*). Previously, *Wei et al. (2019)* found that the K ion content of the 'Korla fragrant pear' and 'Xinli 6' were 1,190 and 960 mg/kg respectively, and *Shi et al. (2022)* found that the K ion content of the 'Huangguan pear' was 1,239.06 mg/kg. The experimental results of this study showed for the first time that the average content of K ions in the peel and pulp of *P. ussuriensis* was 1,838.06 and 1,615.81 mg/kg, respectively, indicating that *P. ussuriensis* is a very valuable, high-K fruit, as the K content values were far higher than in the pear varieties previously studied.

Calcium ion content was the second largest, followed by phosphorus and magnesium ion. In the fruit pulp, the P content was second only to K content, followed by Mg and Ca content, which is consistent with the research results and conclusions of *Aizezi et al. (2018)* on the 'Korla fragrant pear.' Ca content was significantly higher in the peel than in the pulp. The ionic radius of Ca is larger than that of magnesium, which leads to greater flow resistance in the phloem, so it has lower fluidity than magnesium (*Karley & White, 2009*). Bivalent Ca ions with larger ionic radii are more conducive to the formation of dense and stable calcium phosphate, calcium carboxylate epidermal plant fiber and phloem protein bone structure with the amino acid residues in the cell structure of the peel and the dense cell membrane carboxylate, which also causes the Ca content to be significantly higher than P and Mg content (*Gu et al., 2022a*, *2022b*). Eating pears without peeling the fruit first

can help supplement the Ca content required by the human body and help maintain the health of both bones and muscles.

Although the contents of trace elements in *P. ussuriensis* were far lower than the main elements, they play important roles in the quality of the pear fruit, a conclusion that has also been reached in the studies of other fruits and vegetables (*Kuang et al., 2022*; *Liu et al., 2010*; *Liu et al., 2017*; *Sardar et al., 2022*). Although *Wang et al. (2021)* and *Zhang et al. (2022b)* found that the trace elements in the 'Laiyang Pear' and 'Hongzaosu Pear' followed a Fe > Cu > Zn content pattern, the results of this study showed that the trace elements in *P. ussuriensis* fruit followed a Fe > Zn > Cu content pattern. This difference in the sequence of trace element content may be related to factors such as variety, tree age, cultivation conditions, and also to the impact of the natural environment such as the soil conditions and climate where the fruit trees grow (*Fan et al., 2012*). *P. ussuriensis* varieties should be selected based on market demand and the conditions of the growing area. The application of calcium and trace element fertilizers should, which can improve the nutritional quality pears, should also be considered.

The results showed that the content of mineral elements in the peel and pulp of *P. ussuriensis* changed in different degrees during their growth, showing rich genetic diversity. In the peel, the largest coefficient of variation was Na, and the smallest coefficient of variation was K. The relevant coefficient of variation ranged from 29.81% to 77.13%. The highest coefficient of variation in pulp was Ca, and the smallest coefficient of variation was Mg. The relevant coefficient of variation ranged from 28.03% to 67.66%. These results showed that the germplasm resources of *P. ussuriensis* had great potential for improvement in mineral element content, and the rich genetic background could provide sufficient parent materials for the breeding of improved varieties.

## Correlation of mineral elements in different parts of *P. ussuriensis* fruit

There is a correlation between the content of mineral elements in peel and pulp. Factor analysis is an effective grouping method based on the correlation between variables that can show the high correlation between the same group of varieties. Although most mineral elements in fruits have positive correlations and synergistic effects, a few of them also have negative correlations and antagonistic effects (*Reddy et al., 2020*; *Yao, Xie & Zeng, 2017*). This study also found this phenomenon. For example, *Duan et al. (2013)* analyzed the mineral quality of 30 pear resource fruits and found that K and Ca were significantly positively correlated in fruits, while this study unexpectedly found that K and Ca were negatively correlated. This abnormal finding may be due to the types of fruits tested, the content of mineral elements in the soil of the sampling area, or the different sampling periods between the two. We also found that there was a significant negative correlation between K and Pb content in the fruit of *P. ussuriensis*, with a correlation coefficient of −0.301. This finding is also contrary to previous research results and conclusions (conventional positive correlation). This abnormal result shows that the Pb content in high-K *P. ussuriensis* varieties is relatively low, indicating that potassium fertilizer application on fruit trees may help reduce harmful lead content in fruit.

There was a significant positive correlation between P and K, P and Cu, and P and Zn in the peel and pulp of the *P. ussuriensis* varieties tested, with correlation coefficients above 0.5. This is consistent with the research results of *Kuang et al. (2017)* on apples and *Guo, Yu & Wu (2019)* on pears. This phenomenon may be related to the synergistic absorption of mineral elements in fruits, as most mineral elements form plant cell structures through chemical or ionic bonds, and promote the biochemical reaction of plant growth through mutual balance and mutual restriction, giving them specific physiological functions.

A factor analysis showed that the key factor affecting the evaluation of mineral elements in the fruit of *P. ussuriensis* was the content of Cu, P, K, Zn, Na and Cr. These research results should provide a valuable reference for the screening and identification of mineral elements in different varieties of *P. ussuriensis*. Previous research results show that the peel is rich in Ca and P, which are essential nutrient for the human body (*Yoshimizu et al., 2015*; *Ayache et al., 2015*), and should be more widely used in food production. These mineral elements can be used as nutritional fortifiers in food additives to supplement and strengthen certain foods (*Liu & Zhao, 2020*; *Sun et al., 2016*), and as tools to regulate and improve food quality. For example, phosphorus can be used as an acidifier in soft drinks (*Li et al., 2015*), and calcium can be added to improve the hardness of canned vegetables (*She, Zhang & Huang, 2014*). *P. ussuriensis* varieties with high calcium and phosphorus content play very important roles in improving the nutritional value of food and protecting human health, and are a great source for research in balancing nutrients in fruits. Although the content of mineral elements in fruit is small, these mineral elements actively participate in many important functions in the human body. Future orchard management plans can include fertilization adjustments based on local conditions and the absorption characteristics of mineral elements in different autumn pear germplasms.

## Cluster analysis of *P. ussuriensis* fruit

In recent years, more researchers have used a cluster analysis to analyze the quality of fruit with multiple samples and indicators (*Qin et al., 2012*; *Yazdanpour, Khadivi & Khah, 2018*). This method has been widely used in the comprehensive evaluation of the quality of plum, peach, grape, apple, orange, as well as other fruits (*Nie et al., 2000*). In the cluster analysis of *P. ussuriensis*, the peel and pulp of 70 *P. ussuriensis* varieties were divided into three categories according to mineral element content. For example, in the correlation analysis of 'ZLTSL-1', the high K and Fe content in the peel placed this variety in Class I of peel groupings, and the high P, Cu and Zn content in the pulp placed this variety into. Class II of pulp groupings. In the correlation analysis of 'HJQ' the high Na, Mg, P and K content in the peel placed this variety into class I of peel groupings, and the high Cr content in the pulp placed this variety into Class III of pulp groupings.

Among the three peel groupings, Class I and Class III included wild cultivated and domesticated varieties, but Class II included only domesticated varieties. Among the three pulp groupings, Class I included both Iwild and domesticated varieties, Class II included both cultivated and domesticated varieties, and Class III included wild, cultivated and domesticated varieties. These results show that while domesticated varieties improve upon some traits of cultivated varieties, such as disease resistance, they do not have significant

advantages in mineral elements content levels. Our comprehensive analysis of the mineral elements of *P. ussuriensis* fruits found that there were significant differences in Mg, P, K, Na and Ca content between the peel and the pulp. Future research can use molecular biology methods to carry out gene mapping and gene mining of *P. ussuriensis* fruit for germplasm resources, aiming for polymorphism of these mineral elements in the fruit, to provide a more in-depth basis for the future genetic improvement of the mineral elements in *P.Ussuriensis* resources.

## CONCLUSIONS

Our research results of *P. ussuriensis* show that content levels of the 12 mineral elements measured were K > P > Ca > Mg > Na > Al > Fe > Zn > Cu > Cr > Pb > Cd, from high to low, in which K and P were the most abundant mineral elements. Different varieties of *P. ussuriensis* can be divided into three categories according to the mineral content of in peel: (1) varieties with high Na, Mg, P, K, Fe and Zn content, (2) varieties with high Ca content, and (3) varieties with medium element content. Different varieties of *P. ussuriensis* can also be divided into three categories according to the mineral content of the pulp: (1) varieties with high Mg, P and K content, (2) varieties with low element content, and (3) varieties with high Na, Ca content.

The comprehensive analysis results found that 'SSHMSL', 'QYL', 'SWSL' and 'ZLTSL-3' are the *P. ussuriensis* varieties with the highest content of mineral elements. The study also found that the application of potassium fertilizer can help reduce harmful lead content in fruit. In addition, ingesting the peel when eating pears is important for supplementing the calcium needed by the human body. These research results are important for the future selection, production and breeding of *P. ussuriensis* fruits.

## ACKNOWLEDGEMENTS

We would like to thank the National Field Genebank for Hardy Fruits (Gong Zhuling City) for providing the fruit tested in this study.

### Funding

This work was supported by the National Modern Agricultural Industry Technology System Supported by Ministry of Finance and Ministry of Agriculture and Rural Affairs (No. CARS-28), the National Natural Science Foundation of Heilongjiang (No. SS2021C003), and the Agricultural Science and Technology Innovation Crossing Project of Heilongjiang Academy of Agricultural Sciences (No. HNK2019CX11). The funders had no role in study design, data collection and analysis, decision to publish, or preparation of the manuscript.

### Grant Disclosures

The following grant information was disclosed by the authors:
National Modern Agricultural Industry Technology System Supported by Ministry of

Finance and Ministry of Agriculture and Rural Affairs: CARS-28.
Natural Science Foundation of Heilongjiang: SS2021C003.
Agricultural Science and Technology Innovation Crossing Project of Heilongjiang
Academy of Agricultural Sciences: HNK2019CX11.

## Competing Interests

The authors declare that they have no competing interests.

## Author Contributions

- Chang Liu conceived and designed the experiments, prepared figures and/or tables, and approved the final draft.
- Honglian Li analyzed the data, prepared figures and/or tables, and approved the final draft.
- Aihua Ren performed the experiments, prepared figures and/or tables, and approved the final draft.
- Guoyou Chen performed the experiments, prepared figures and/or tables, and approved the final draft.
- Wanjun Ye performed the experiments, prepared figures and/or tables, and approved the final draft.
- Yuxia Wu analyzed the data, authored or reviewed drafts of the article, and approved the final draft.
- Ping Ma performed the experiments, prepared figures and/or tables, and approved the final draft.
- Wenquan Yu conceived and designed the experiments, authored or reviewed drafts of the article, and approved the final draft.
- Tianming He conceived and designed the experiments, authored or reviewed drafts of the article, and approved the final draft.

## Data Availability

The original measurements are available in the Supplemental File.

## Supplemental Information

Supplemental information for this article can be found online at http://dx.doi.org/10.7717/peerj.15328#supplemental-information.

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
