# Peer review of "A comparison of the mineral element content of 70 different varieties of pear fruit (Pyrus ussuriensis) in China"

_PeerJ, doi:10.7717/peerj.15328_

## Round 0.1 · original submission · Minor Revisions

Authors are advised to revise the manuscript as per comments and suggestions of the reviewers.

Reviewer 1 ·

Basic reporting

• What was the specific rationale to perform this study because several authors have reported this type of data already?

Experimental design

What was the criteria for selecting the no. of samples with respect to the various locations?

Validity of the findings

Results are not written properly. The authors should try to write concisely.

Additional comments

I have thoroughly reviewed the manuscript and suggested major revisions, mostly in the results and discussion that need to be clearly addressed by the authors before I can recommend the publication of the manuscript. My comments are as the following:
There are some general and specific comments:
• The work is quite interesting but the main innovative idea is missing.
• Moderate linguistic (English) corrections are desirable. In addition, the submitted version has numerous sentence framing/ phrase construction issues.
• Novelty of research work is also missing. Please add future recommendation.
• Some of the paragraphs are needs to be rearrange for better clarification.
• Avoid repeating the same information.
• Please improve discussion through some linking paragraphs and recent literature. This section is too lengthy.
• Please highlight the main or interesting findings for each parameter.
• Please include the clear take home message.

·

Basic reporting

no comments

Experimental design

no comments

Validity of the findings

no comments

·

Basic reporting

Self-contained with relevant results to hypotheses

Experimental design

Methods described with sufficient detail & information to replicate.

Validity of the findings

All underlying data have been provided; they are robust, statistically sound, & controlled.

Additional comments

In the manuscript “Factor analysis and cluster analysis of mineral elements contents of Pyrus ussuriensis Maxim. at maturity stage in Northeast China”, the authors have work hard to execute the current study”. Title of the manuscript is interesting and covers the scope of the journal. Nevertheless, the manuscript may be a suitable candidate for publication after the following amendments:
• Please add some necessary details to the background information in the abstract.
• Add name of some bioactive substances in Line 67.
• Figure 1 and 2 lacking standard error bar.
• Please subscript the P<0.01.
• Briefly write some future perspective in abstract.
• There are some typos and grammatical mistakes in the abstract and rest of the sections of the manuscript, please improve the language of the manuscript.
• Please write the keywords alphabetically. Don’t write those words as a keyword which is a part of manuscript’s title.
• What is novelty of the current study?
• Please clearly write hypothesis and objectives of the current study.
• What are the future perspectives of this study?
• Please give detail about statistic design and sampling size used during the current study.
• Please add further logical discussion about results obtained.
• Recheck all typos and improve overall formatting used in MS.
• References are not enough and up to date. Beside they are also not to the journal format.
• Conclusion looks like as an abstract. Rewrite the conclusion for better clarity.

·

Basic reporting

This manuscript is well-written and contains concise information. Well-understanding information of manuscript for the reader present. All the references are relevant in this manuscript. But for more clarity reader cite more authentic literature. All the figures, and tables well organized and completely explained in the manuscript. Results are well-defined.

Experimental design

Research conducted to describe the issue, methodology is well defined. According to the title and journal policy research gap is filled by using modern techniques. collected information using the well precise methodology. The write-up of this paper is well structured and easy to understand for readers.

Validity of the findings

However, according to my opinion provided information for easy understanding
should be more descriptive. I suggested explaining the objectives of this research article with
more recent literature. In my opinion manuscript information for the reader
understanding, further, described in more depth. The overall theme of the manuscript is well
clear. Moreover, this is good quality research work, more its practical implication
in future perspective is needed for industrial purposes.

Additional comments

Line No. 141: Describe solution preparation reference.
Line No. 159: Describe for clear understanding blank solution of sample preparation.

---

## Round 0.2 · accepted · Accept

Authors have revised the manuscript as per suggestions. Therefore, I recommend that manuscript can be accepted for publication.

·

Basic reporting

The authors have revised the manuscript as per suggestions. Therefore, I recommend that manuscript can be accepted for publication.

Experimental design

The authors have revised the manuscript as per suggestions. Therefore, I recommend that manuscript can be accepted for publication.

Validity of the findings

The authors have revised the manuscript as per suggestions. Therefore, I recommend that manuscript can be accepted for publication.

Additional comments

The authors have revised the manuscript as per suggestions. Therefore, I recommend that manuscript can be accepted for publication.

·

Basic reporting

Now the English language has been improved overall. All the concepts are clear and easy to understand.

Experimental design

After re-review, there has been no addition required. No further changes are required.

Validity of the findings

no comments. Findings are acceptable